# Adherence and Toxicity during the Treatment of Latent Tuberculous Infection in a Referral Center in Spain

**DOI:** 10.3390/tropicalmed8070373

**Published:** 2023-07-19

**Authors:** Juan David Puyana Ortiz, Andrea Carolina Garcés Rodríguez, María Luisa Aznar, Juan Espinosa Pereiro, Adrián Sánchez-Montalvá, Joan Martínez-Campreciós, Nuria Saborit, José Ángel Rodrigo-Pendás, Guadalupe García Salgado, Claudia Broto Cortes, Nuria Serre Delcor, Inés Oliveira, Begoña Treviño Maruri, Diana Pou Ciruelo, Fernando Salvador, Pau Bosch-Nicolau, Irene Torrecilla-Martínez, Ricardo Zules-Oña, María Teresa Tórtola Fernández, Israel Molina

**Affiliations:** 1International Health and Cooperation UAB, 08193 Barcelona, Spain; 2Morales Meseguer University Hospital, 30008 Murcia, Spain; 3International Health Unit Vall d’Hebron-Drassanes, Infectious Diseases Department, Vall d’Hebron University Hospital, PROSICS Barcelona, 08001 Barcelona, Spaind.pou@vhebron.net (D.P.C.); fernando.salvador@vallhebron.cat (F.S.);; 4Centro de Investigación Biomédica en Red de Enfermedades Infecciosas (CIBERINFEC), Instituto de Salud Carlos III, 28029 Madrid, Spain; 5Grupo de Estudio de Micobacterias (GEIM), Sociedad Española de Enfermedades Infecciosas y Microbiología Clínica (SEIMC), 28003 Madrid, Spain; 6Preventive Medicine and Epidemiology Department, Vall d’Hebron University Hospital, 08035 Barcelona, Spain; 7Microbiology Department, Vall d’Hebron University Hospital, 08035 Barcelona, Spain

**Keywords:** *Mycobacterium tuberculosis*, tuberculosis screening, latent tuberculosis infection, adverse events, toxicity, adherence

## Abstract

The screening and treatment of latent tuberculosis infection (LTBI) in countries with a low incidence of TB is a key strategy for the elimination of tuberculosis (TB). However, treatment can result in adverse events (AEs) and have poor adherence. This study aimed to describe treatment outcomes and AEs for LTBI patients at two departments in Vall d’Hebron University Hospital in Barcelona, Spain. A retrospective study was conducted on all persons treated for LTBI between January 2018 and December 2020. Variables collected included demographics, the reason for LTBI screening and treatment initiation, AEs related to treatment, and treatment outcome. Out of 261 persons who initiated LTBI treatment, 145 (55.6%) were men, with a median age of 42.1 years. The indications for LTBI screening were household contact of a TB case in 96 (36.8%) persons, immunosuppressive treatment in 84 (32.2%), and recently arrived migrants from a country with high TB incidence in 81 (31.0%). Sixty-three (24.1%) persons presented at least one AE during treatment, and seven (2.7%) required definitive discontinuation of treatment. In the multivariate analysis, AE development was more frequent in those who started LTBI treatment due to immunosuppression. Overall, 226 (86.6%) completed treatment successfully. We concluded that LTBI screening and treatment groups had different risks for adverse events and treatment outcomes. Persons receiving immunosuppressive treatment were at higher risk of developing AEs, and recently arrived immigrants from countries with a high incidence of TB had greater LTFU. A person-centered adherence and AE management plan is recommended.

## 1. Introduction

Tuberculosis (TB) is still a major public health problem. It has been estimated that a quarter of the world’s population is infected by *Mycobacterium tuberculosis* [1]. Latent tuberculosis infection (LTBI) is defined as a status of infection control due to a persistent immune response generated by the antigens of the *M. tuberculosis* complex without evidence of clinical manifestations of active TB. Without treatment, between 5 and 10% of people with LTBI will develop active TB at some point in their lives [2]. In 2015, the World Health Organization (WHO) launched the End TB Strategy, with the targets of 90% reduction in individuals suffering from TB and 95% reduction in TB deaths by 2035. The detection and treatment of people with LTBI with a higher risk of progression to active TB is deemed as a key component of this strategy [3,4].

Screening for LTBI should be performed or considered depending on the underlying epidemiological burden in specific at-risk populations, such as prisoners, healthcare workers, immigrants from countries with a high TB burden, homeless individuals, and people who use illicit drugs (Appendix A, Table A1) [3,4].

Since it is not possible to sample and culture dormant mycobacteria in those chronically infected without symptoms, there is no gold standard test for LTBI. Our only means of diagnosing LTBI is to prove a persistent immune response against *M. tuberculosis* antigens. Two types of tests are approved for the diagnosis of LTBI, the tuberculin skin test (TST) and the IFN-gamma release test (IGRA). WHO guidelines recommend the use of either of these two tests for the diagnosis of LTBI, depending on the availability and affordability of the tests [3,4].

The main objective of LTBI treatment is to prevent its progression to active TB. There are several treatment regimens recommended, and the selection of treatment should consider the characteristics of the individuals and concomitant medication to avoid possible toxicities or interactions [3,4,5]. Completion of treatment is critical for success both at the individual and at the programmatic level. However, several studies have yielded different results regarding treatment completion rates that vary according to the risk group [6,7,8]. Adverse events (AEs) related to medication may be one of the main factors related to poor treatment completion. In fact, the fear of developing an AE is the main reason for not offering LTBI treatment to recently arrived migrants older than 35 years old [9,10,11,12,13,14]. While most of these reactions are minor and rarely occur, specific attention should be paid to preventing drug-induced hepatotoxicity. 

The WHO provides clear instructions on the follow-up of individuals receiving treatment for LTBI, recommending monthly check-ups by healthcare providers. However, some of the interventions lack evidence, such as testing of baseline liver function or if there are incremental benefits of routine monitoring of liver enzyme levels over education and clinical observation alone for preventing severe clinical AE [15]. 

The aim of this work is to describe the risk factors associated with adherence and AE in persons who started LTBI treatment.

## 2. Materials and Methods

### 2.1. Study Design

This is a retrospective study carried out at the Tropical Medicine and International Health Unit Vall d’Hebron-Drassanes and at the Preventive Medicine and Epidemiology Department of the Vall d’Hebron Barcelona Hospital Campus in Barcelona, Spain. It is a tertiary care center that covers a health area with a population of more than 430,000 people. Additionally, it is a national referral center for tropical diseases and tuberculosis, among other conditions [16].

The inclusion criteria are people 18 years of age or older who started treatment for LTBI between January 2018 and December 2020. Participants living with HIV or who were going to receive transplants were excluded from the study because treatment monitoring and follow-up of these patients is performed by another medical department.

### 2.2. LTBI Screening 

Screening for LTBI was performed using either IGRA (LIAISON·QuantiFERON TB·Gold·plus·DiaSorin, Saluggia, Italy) or TST. When TST was used, a skin induration measuring 15 mm or longer was considered a positive result, assuming that immigrants coming from countries with a high burden of TB have been vaccinated with BCG. The reason for LTBI screening was classified into three categories: Persons receiving immunosuppressive therapy for oncological and/or autoimmune disease treatment, household contacts of patients diagnosed with pulmonary TB, and recently arrived (<5 years) immigrants from areas with a high incidence of TB (incidence > 100 cases/100,000 inhabitants-year).

### 2.3. Variables of Interest

Demographics (age, sex, country of origin, year of arrival if migration), clinical characteristics (comorbidities, concomitant treatment), blood test parameters, AEs, treatment adherence, and treatment outcome were collected through the electronic health record (EHR). Blood tests included complete blood count and biochemistry, including creatinine, aspartate aminotransferase (AST), alanine aminotransferase (ALT), gamma glutamyl transpeptidase, total bilirubin, direct bilirubin, and alkaline phosphatase. The schemes used in our unit for the treatment of LTBI are Rifampicin and isoniazid for 3 months (3RIF/INH), isoniazid for 6 months (6INH) or rifampicin for 4 months (4RIF). Follow-up is usually performed by nurses specialized in the management of these treatments and is usually performed two weeks after starting treatment, then after a month and a half and then every 2 months.

Treatment outcomes were treatment complete or treatment incomplete for any reason, including low adherence, lost to follow-up (LTFU), or discontinuation due to toxicity. Persons were considered to complete treatment if they completed more than 80% of the pre-specified treatment duration. Termination of treatment due to toxicity was defined as all those persons who discontinued treatment for more than one month associated with any AEs. We did not include post-treatment follow-up to assess the number of participants who developed active TB.

AEs were classified into cutaneous symptoms, gastrointestinal symptoms, neurological symptoms, and others. The severity of AEs was defined based on the Division of Microbiology and Infectious Diseases (Division of AIDS (DAIDS) Table for Grading the Severity of Adult and Pediatric Adverse Events, Version 2.1 (Appendix A, Table A2 and Table A3). Serious AEs were defined as (a) AEs that caused treatment discontinuation; (b) AE that caused change of treatment; and (c) any AEs classified as grade 3 or 4 of DMID.

### 2.4. Statistical Analysis

The quantitative variables were described using the mean and the standard deviation (SD) if they were normally distributed or the median and the interquartile range (IQR) if they were not. Categorical variables were described by frequencies (n) and percentages (%). 

A bivariate analysis was performed to determine risk factors for AE development (presence or absence of AE), serious AEs (presence or absence of serious AEs), and treatment outcome (complete treatment vs. treatment not completed). To establish comparisons between groups, the Student’s *t* test was used if the quantitative variable had a normal distribution, and the Mann–Whitney U test was used otherwise. To compare categorical variables, the Chi-squared test or Fisher’s exact test was used, as appropriate. To assess the association between the different characteristics of the individuals and the adverse events, the Odds Ratio (OR) was calculated with its 95% confidence interval (95% CI). 

A multivariate analysis was carried out using a logistic regression model that included clinically relevant variables and those with a *p*-value < 0.2 in the bivariate analysis.

The bivariate and multivariate analyses were carried out, (a) including all patients and (b) excluding those patients who were LTFU. Variables with a *p*-value < 0.20 in the bivariate analysis or with *p* > 0.20 but considered clinically significant upon consensus between the authors were included in the multivariate analysis. Results with a *p*-value < 0.05 were considered statistically significant.

The data were obtained from electronic clinical records, and all the variables were collected in a database created in the REDCap program (Vanderbilt University, Nashville, TN, USA). Statistical analysis was performed with SPSS statistics v2019.

### 2.5. Ethics

The study followed the tenets of the Declaration of Helsinki and was approved by the institutional review board. Patient consent was waived due to the retrospective nature of the study. 

## 3. Results

A total of 261 persons were included; 145 (55.6%) were males, and median (IQR) age was 42.1 (28.9–58.9) years. The indications for LTBI screening were as follows: household contact with a TB case in 96 (36.8%) persons, immunosuppressive treatment in 84 (32.2%), and recently arrived migrants from a country with high TB incidence in 81 (31.0%). 

When analyzing basal characteristics variables according to the reason for LTBI screening, we observed that all three groups significantly differed in sex proportions, age, and comorbidities. The main baseline characteristics according to the indication for LTBI screening and treatment are summarized in Table 1.

The initial treatment for LTBI was 3RIF/INH in 157 (60.2%) persons, 6INH in 93 (35.6%) persons, and 4RIF in 11 (4.2%) persons. The combination of 3RIF/INH was more common among recently arrived migrants and household contacts compared with immunosuppressed persons (98.8% vs. 69.8 % vs. 11.9%, respectively, *p* < 0.001).

### 3.1. Adverse Events

Overall, 108 (41.4%) participants developed at least one AE. A total of 57(21.8%) developed clinical symptoms related to medication: gastrointestinal toxicity in 42 (16.1%), dermatological toxicity in 6 (2.3%), neurological toxicity in 4 (1.5%), and other clinical symptoms in 5 (1.9%). Hepatotoxicity was observed in 64 (24.5%) participants, with 54 (84.3%) of them being grade I or II, and neutropenia was observed in 8 (11.8%), with 6 (75.0%) of them being grade I or II. AEs were observed in 49 (58.3%) persons receiving immunosuppressive treatment, 39 (40.6%) household contacts, and in 20 (24.7%) recently arrived migrants. The mean number of days to develop an AE was 37.9 (29.7) days.

Nineteen (7.3%) persons developed a severe AE. Severe AEs were observed in one (1.2%) recently arrived migrant who had to stop treatment due to gastrointestinal symptoms without abnormalities in blood tests; seven (8.4%) immunosuppressed persons (three due to grade III-IV hepatic toxicity associated with gastrointestinal symptoms, two due to asymptomatic grade III-IV hepatic toxicity, one due to cutaneous toxicity, and one due to gastrointestinal symptoms with no laboratory abnormalities) and eleven (11.5%) household contacts (five due to clinical symptoms without blood test abnormalities, three due to asymptomatic grade III-IV hepatic toxicity, and two due to grade III-IV hepatic toxicity associated with symptoms and one with gastrointestinal symptoms who was lost to follow up after 3 weeks). Dose adjustment or temporary drug interruption was required in 12 (4.6%) persons, and permanent drug interruption was required in 7 (2.7%) persons.

Main AEs according to the reason for LTBI screening and treatment are summarized in Table 2.

Out of 68 persons who presented abnormalities in the blood test, only 17 (25%) had clinical symptoms. On the other hand, out of 57 individuals who developed clinical symptoms, only 17 (29.8%) presented abnormalities in blood test. We did not observe any relationship between symptoms related to toxicity and blood test abnormalities (*p* = 0.379). 

A total of 12/241 (5.0%) presented grade III or IV blood test abnormalities, and only 5 of them (41.7%) presented clinical symptoms. None of the persons who developed severe blood test abnormalities were recently arrived migrants. 

In the bivariate analysis, we observed that being female, being older, a regimen containing 6INH, the immunosuppression of or being in contact with a TB case were the main causes of being treated for LTBI, and having an abnormal baseline liver profile and receiving concomitant medication were associated with the development of AE. However, in the multivariate analysis, only the reason for LTBI treatment was associated with the development of an AE: both immunosuppressed persons and TB contacts were more likely to develop an AE (7.36 (2.04–26.58), *p* = 0.002 and 2.38 (1.08–5.30), *p* = 0.031, respectively). When we exclude persons who were LTFU, immunosuppressed persons continue to be more likely to develop an AE (OR 8.25 (IC 95% 2.15–31.67), *p* = 0.002) (Table 3 and Table 4).

### 3.2. Treatment Adherence

Overall, 229 (87.7%) persons completed LTBI treatment, 25 (9.6%) were LTFU, and 7 (2.7%) had to stop the treatment due to AEs. 

We observed that people who were LTFU were younger (36.8 (18.6) years vs. 45.8 (18.4) years, *p*= 0.021). We also observed differences between patients who completed treatment and those who were LTFU regarding the reason for LTBI treatment and LTBI regimen. The main variables of persons who completed treatment and those who were LTFU are summarized in Table 5. 

## 4. Discussion

In our study, we analyzed three very different groups: (1) migrants from countries with high TB incidence, who were mainly males and significantly younger and healthier than the others; (2) persons who needed IS treatment, who were more frequently female and older and with more comorbidities than the others; and (3) a third group composed of close contacts with a patient with a pulmonary TB index, who were middle-aged and only one-third of which had comorbidities. By design, our study exclude children and adolescents, who are known to be at a particular risk of developing active TB after contact with a pulmonary TB case, as was shown by old studies and confirmed during the COVID-19 lockdown. Groups with different baseline characteristics may therefore require different safety and adherence-monitoring strategies to ensure optimal treatment outcomes.

There are four approved treatment schemes for LTBI: 6INH, 4RIF, 3RIF/INH, and weekly rifapentine associated with isoniazid for 3 months. The latter is not currently available in Spain. He 3RIF/INH scheme was the most used in our study (60.2%), being applied mainly in newly arrived immigrants and in those with TB household contacts. Possible drug–drug interaction between RIF and IS treatments among people who were diagnosed with LTBI due to IS might be the reason for this lower use of the 3RIF/INH regimen in this group. Interestingly, we did not observe statistically significant differences in terms of the development of AEs between different regimens. This result, together with the finding that severe adverse events were more frequent in the IS than in recently arrived migrants, suggests that baseline characteristics rather than treatment schedules are key to establishing the risk of toxicity. 

Overall, 41.4% of the persons in our study had some type of AE. Gastrointestinal symptoms were the most prevalent, followed by skin and neurological symptoms; these clinical symptoms were very similar to the ones reported in the literature [11]. In other series in which LTBI has been treated with INH, AE percentages of up to 5.4% have been observed [17,18]. These differences could be due to different population characteristics or different definitions of the AE used in the other studies.

Regarding alterations in blood tests, 64 (24.5%) persons showed any grade of liver enzyme elevations. However, hepatic alteration prevailed within grades I and II in most of the persons. It is true that most patients were receiving concomitant medication that might be the cause of the elevation of liver enzymes, so we cannot ensure that hepatotoxicity was due to LTBI treatment. Only 10 (3.8%) persons presented severe liver enzyme abnormalities (grade III and IV). Interestingly, only five of them presented concomitant clinical symptoms, and none of them were recently arrived immigrants. 

In other studies, clinical hepatitis (elevation of liver enzymes accompanied by clinical symptoms) rates can range between 0% and 2.4%, occurring mostly in the first 3 months of treatment [19]. In general, the latest WHO guidelines recommend clinical monitoring of persons undergoing treatment for LTBI and only perform laboratory tests in persons with alterations in the baseline analysis or in persons who develop clinical symptoms during treatment [3]. If these guidelines have been followed, people with severe laboratory abnormalities who did not develop clinical symptoms would not have been found. This fact suggests that blood analysis during follow-up might be necessary in certain risk groups of persons to identify early signs of toxicity without the presence of clinical symptoms [14].

We observed that immunosuppressed persons and TB contacts were more likely to develop an AE compared to recently arrived migrants. A potential explanation is that recently arrived migrants are younger and healthy persons with no comorbidities and no concomitant medication. In contrast, immunosuppressed individuals are subjected to polypharmacy (with hepatic metabolism) and suffer from various comorbidities, which increases the risk of AEs. Moreover, in our series, TB contacts also had a higher number of comorbidities compared to newly arrived migrants. This leads us to consider a closer follow-up in certain cases that require special immunosuppressive treatments or other comorbidities [20,21,22,23,24,25,26].

It is striking that, in our study, we did not observe statistically significant differences in age between people who developed AE and those who did not develop AE. Several previous studies have observed advanced age as a risk factor for developing AEs [27,28,29]. However, other studies have also failed to demonstrate a relationship between age and the development of AEs [30,31]. This finding may be related to the fact that it might not be only age that is related to the development of AEs, but rather a combination of factors associated with age, such as comorbidities or the use of concomitant drugs. In this sense, it might be worthwhile to treat LTBI in newly arrived immigrants regardless of age since the main reason for not doing so is the risk of hepatotoxicity associated with the treatment.

Compliance with and adherence to LTBI treatment play a very important role in the goal of reducing and eliminating active TB worldwide, which is why emphasis is placed on programs to monitor persons who receive such treatment [3,4]. Different studies refer to adherence to LTBI treatment ranging between 22% (95% CI: 6–43%) and 82% (95% CI: 66–94%) depending on the risk group the persons belong to [6,7,8]. It has been found that in prisoners and immigrants, the rate of adherence is lower compared to TB contacts and persons living with HIV [6,32]. In another study from the southern cone in South America, adherence to treatment for LTBI was 85.3%, and LTFU represented 9.2% [33].

In our study, we observed a high rate of treatment completion, represented by 86.7% of persons. However, we observed differences in treatment outcomes regarding the reason for LTBI treatment, namely that recently arrived migrants were more likely to receive LTFU LTBI treatment. The causes of LTFU in this group of persons might be multifactorial, such as language barriers, migration conditions, cultural conception of disease-medical care, and idiosyncrasies/beliefs/cultural traits that can lead to treatment abandonment [33,34]. Moreover, migrants with poor social backup move more in search of jobs. In the setting of a wide public health network such as in Spain, one approach could be to establish a coordinated care between different health areas, allowing people to move, searching for job opportunities while retaining them on treatment. In addition, immigrant people coming from some countries, such as the Sub-Saharan region, used to be afraid of and mistrust blood extraction [30]. This fact might be another cause of LTFU in the case that blood tests would be routinely performed. We consider that it will be important in the future to design tailored-group strategies that approach the reasons behind poor adherence in every group. For this reason, based on the findings of our study, where recently arrived migrants were persons with fewer comorbidities, fewer AEs to treatment, younger age group, and greater LTFU, recommending therapy with limited hematological follow-up may be an option to improve adherence to therapy.

There are several limitations in our study. First, it is a retrospective study, so there are intrinsic methodological limitations. For instance, although we found a high incidence of AE of any grade, clinicians may not register the mildest symptoms. Second, it has been performed in a single center, so conclusions may not be reproducible in other contexts. Third, children, people living with HIV, and transplant candidates were not included, so conclusions in this group of persons might not be drawn. Fourth, it is difficult to differentiate whether hepatotoxicity is due to LTBI treatment or to immunosuppressive treatments in the case of immunosuppressed individuals, because both treatments are usually started at the same time.

## 5. Conclusions

In our study, the indication of LTBI screening and treatment defined groups with different sociodemographic and clinical characteristics, which defined different risks for AE and treatment outcomes. We observed a higher risk of developing AEs in the group of persons receiving IS treatment and a lower adherence to treatment in recently arrived immigrants from countries with a high incidence of TB. Strategies should be strengthened to increase adherence to treatment in recently arrived immigrants and to ensure comprehensive follow-up of AE in those with baseline IS.

## Figures and Tables

**Table 1 tropicalmed-08-00373-t001:** Baseline characteristics in persons who started treatment for LTBI.

	Recently Arrived Migrants N = 81	Immunosuppressed Persons N = 84	Household Contacts N = 96	*p* Value
Sex, male	57 (70.4)	38 (45.2)	50 (52.6)	0.004
Median age (years), IQR	27.0 (22.7–32.8)	62.4 (49.6–72.4)	44.9 (32.3–55.0)	<0.001
Country of birth				
Spain	0	63 (79.7)	39 (40.6)	<0.001
Comorbidities	6 (7.4)	84 (100)	30 (31.3)	0.001
High blood pressure	0	25 (29.8)	11 (11.5)	<0.001
DM	0	12 (14.3)	6 (6.3)	<0.001
Dyslipidemia	1 (1.2)	20 (23.8)	10 (10.4)	<0.001
Neoplasia	0	45 (53.6)	4 (4.2)	<0.001
Autoimmune disease	0	41 (48.8)	6 (6.3)	<0.001
Other	6 (7.4)	37 (44.0)	24 (25.0)	<0.001
Concomitant treatment *	1 (1.2)	76 (90.5)	23 (24.0)	<0.001
Treatment regimen for LTBI				
3RIF/INH	80 (98.8)	10 (11.9)	67 (69.8)	<0.001
6INH	1 (1.2)	74 (88.1)	18 (18.8)	
4RIF			11 (11.5)	

* Concomitant treatment at the moment of LTBI treatment initiation. There were patients with more than one comorbidity. IQR: Interquartile Range, DM: Diabetes mellitus; 3RIF/INH: 3 months of rifampin and isoniazid; 6INH: 6 months of isoniazid; 4RIF: 4 months of rifampin.

**Table 2 tropicalmed-08-00373-t002:** Main adverse events (AEs) according to the reason for LTBI screening and treatment.

	Recently Arrived Migrants	Immunosuppressed Persons	Household Contacts	*p*-Value
	N = 81	N = 84	N = 96	
Any AEs	20 (24.7%)	49 (58.3%)	39 (40.6%)	<0.001
Gastrointestinal disorders	7 (8.6%)	22 (26.2%)	13 (13.5%)	0.006
Dermatological disorders	2 (2.5%)	2 (2.4%)	2 (2.1%)	0.984
Neurological disorders	0	2 (2.4%)	2 (2.1%)	0.396
Other	1 (1.2)	1 (1.2)	3 (3.1)	0.554
Hepatic toxicity	10 (14.3)	29 (36.7)	25 (27.8)	0.008
I	8	19	15	
II	2	5	5	
III	0	2	4	
IV	0	3	1	
Neutropenia	2 (2.5)	5 (6.0)	1 (1.0)	0.256
I	2	1	1	
II	0	2	0	
III	0	2	0	
Severe AE	1 (1.2)	7 (8.4)	11 (11.5)	0.03
Treatment change	2 (2.5)	2 (2.5)	8 (8.3)	0.089
Treatment discontinuation	1 (1.2)	5 (6.0)	1 (1.0)	0.079

**Table 3 tropicalmed-08-00373-t003:** Variables related to the development of AE (bivariate and multivariate analysis). All participants.

	All Participants					
			Bivariate		Multivariate	
	No AEs N = 153	AEs N = 108	OR (95%CI)	*p* value	OR (95%CI)	*p* value
Sex, Female	58 (38.2)	57(58.8)	1.81 (1.10–2.99)	0.020	1.67 (0.94–2.82)	0.082
Age, mean (SD)	42.7 (18.8)	48.4 (18.0)	1.02 (1.00–1.03)	0.016	0.99 (0.97–1.02)	0.580
LTBI regimen						
3RIF/INH **	104 (68%)	53 (49.1)	1 (REF)		1 (REF)	
6INH	44 (28.8)	49 (45.4)	2.18 (1.29–3.69)	0.003	0.79 (0.33–1.88)	0.594
4RIF	5 (3.3)	6 (5.6)	2.35 (0.69–8.07)	0.173	1.41 (0.38–5.27)	0.611
Reason for LTBI treatment						
Recently arrived migrants *	35 (22.9)	20 (18.5)	1 (REF)		1 (REF)	
Immunosuppressed	57 (37.3)	49 (45.4)	4.27 (2.19–8.31)	<0.001	7.36 (2.04–26.58)	0.002
Contact	9 (6.1)	39 (36.1)	2.09 (1.09–3.99)	0.026	2.38 (1.08–5.30)	0.031
Baseline abnormal hepatic profile		14 (13.6)	2.43 (1.01–5.85)	0.048	2.28 (0.89–5.82)	0.084
Concomitant medication	48 (31.4)	52 (48.1)	2.03 (1.22–3.34	0.006	1.30 (0.52–3.21)	0.573

SD: Standard Deviation, INH: Isoniazid, RIF: Rifampin, AE: Adverse Events. * Reference group: recently arrived migrants; ** Reference group: 3RIF/INH.

**Table 4 tropicalmed-08-00373-t004:** Variables related to the development of AE (bivariate and multivariate analysis). Excluding persons LTFU.

	Excluding Persons LTFU					
			Bivariate		Multivariate	
	No AEs N = 137	AEs N = 99	OR (95%CI)	*p* value	OR (95%CI)	*p* value
Sex, female	53 (39.0)	52 (52.5)	1.73 (1.03–2.92)	40	1.58 (0.89–2.82)	119
Age, mean (SD)	44.4 (18.9)	47.9 (17.7)	1.01 (0.99–1.03)	123	0.99 (0.96–1.01)	241
LTBI regimen						
3RIF/INH **	43 (31.4)	46 (46.5)	1 (REF)		1 (REF)	
6INH	89 (65.0)	47 (47.0)	2.03 (1.17–3.50)	11	0.77 (0.31–1.89)	564
4RIF	5 (3.6)	6 (6.1)	2.27 (0.66–7.84)	194	1.56 (0.41–5.93)	513
Reason for LTBI treatment						
Recently arrived migrants *	47 (34.3)	17 (17.2)	1 (REF)		1 (REF)	
Immunosuppressed	34 (24.8)	47 (47.5)	3.82 (1.88–7.76)	<0.001	8.25 (2.15–31.67)	2
Contact	56 (40.9)	35 (35.4)	1.73 (0.86–3.47)	124	2.19 (0.95–5.06)	67
Baseline abnormal hepatic profile	8 (6.0)	14 (14.7)	2.73 (1.10–6.81)	31	2.53 (0.96–6.65)	60
Concomitant medication	47 (34.3)	49 (49.0)	1.88 (1.11–3.18)	20	1.19 (0.46–3.03)	718

* Reference group: recently arrived migrants; ** Reference group: 3RIF/INH.

**Table 5 tropicalmed-08-00373-t005:** Main characteristics of persons who completed treatment and persons who were LTFU.

	Treatment Completed N = 229	LTFU N = 25	*p* Value
Sex, female	102 (44.7)	10 (40.0)	0.651
Age, mean (SD)	45.8 (18.4)	36.8 (18.6)	0.021
Reason for LTBI treatment			
Recently arrived migrants	76 (33.2)	3 (12.0)	<0.001
Immunosuppressed	90 (39.3)	5 (20.0)	
Contact	63 (27.5)	17 (68.0)	
LTBI regimen			
6INH	84 (36.7)	4 (16.0)	0.041
3RIF/INH	134 (58.5)	21 (84.0)	
4RIF	11 (4.8)	0	
AE	92 (40.2)	9 (36.0)	0.686

SD: Standard Deviation, INH: Isoniazid, RIF: Rifampin, AE: Adverse Events.

## Data Availability

No new data were created or analyzed in this study. Data sharing is not applicable to this article.

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
