# Peer review of "Adherence and Toxicity during the Treatment of Latent Tuberculous Infection in a Referral Center in Spain"

_tropicalmed, 2023, doi:10.3390/tropicalmed8070373_

Round 1
Reviewer 1 Report
The concept of this manuscript is appreciated, where the authors describe the risk factors associated with adherence and adverse events in individuals who initiated treatment for latent tuberculosis infection (LTBI). The objective of this study was to provide a comprehensive description of the treatment outcomes and adverse events experienced by patients with LTBI at two departments within Vall d'Hebron University Hospital in Barcelona, Spain. The authors conducted a retrospective analysis, including all individuals who underwent LTBI treatment from January 2018 to December 2020. The study encompassed all individuals who underwent treatment for LTBI during that time period.
Here are some comments for improvement:
Q) It's not clear how immigrants were specifically tested for LTBI screening. Was it through IGRA or TST? If immigrants are from high-incidence demographics, then TST might not be a suitable screening test.
Q) The presence of Table 2 continuation between line 223 and 224, apart from Table 8, might confuse the reader.
Q) Tables 1, 2, and 3 could be shifted to supplementary documents.
Author Response
Q1. Regarding LTBI screening in immigrants, it was performed either by IGRA or TST according to the criteria of the attending physician. When TST was used, a skin induration measuring 15 mm or longer was considered a positive result, assuming that immigrants coming from countries with a high burden of TB have been vaccinated with BCG. We have added this explanatory sentence to the article.
Q2. We have modified this table to make it easier to understand.
Q3. Tables 1, 2 and 3 have been shifted to supplementary documents.
Thank you very much for your careful review of our manuscript.

Reviewer 2 Report
1. Is the BCG vaccine history of the study participants known? Will including this information to the analysis have any benefit to the study?
2. Lines 167-168 "The combination of 3RIF/INH..." is not reflected in the Table 4
3. LTFU- please expand where used first (line 112) and may be removed from line 217
Author Response
Q1. We thank reviewer two for this comment. Unfortunately, the vaccination status of the participants was not recorded in the medical record, so we do not have this information. However, it was assumed that all newly arrived immigrants from countries with a high TB burden had received BCG vaccination, so in the case of TST, we always considered positive values those with skin induration greater than 15 mm, so we believe that the results would not have changed much.
Q2. The lines in table 4 had been moved, so it seemed that this information was not reflected, but it is. This table has been corrected.
Q3. LTFU has been defined in line 112 (used first) and has been removed from line 217.
Thank you very much for your careful review of our manuscript.

Reviewer 3 Report
Dear colleagues,
In the article colleagues have introduced actual data.
The following items should be corrected:
- The keywords should be more than five.
- TB epidemiology data based on the WHO Reports 2020- 2022 should be included in the introduction
- It could be better to update the references. Currently, we have new guideline (WHO consolidated guidelines on tuberculosis. Module 3: diagnosis. Tests for tuberculosis infection. Geneva: World Health Organization; 2022. Licence: CC BY-NC-SA 3.0 IGO.)
- It will be very good if authors include the definition of LTBI as well.
- Characteristics in persons (Table 4 and 8) should be presented in the chapter "Materials"
- Please formulate the conclusions more clearly and precisely with practical relevance clarification. What person characteristics and comorbidities spectrum of patients should be used in LTBI management?
Many thanks for the results!
Author Response
Q1: We have added the following keywords: Mycobacterium tuberculosis, tuberculosis screening.
Q 2,3: We have updated it.
Q4. Latent tuberculosis infection is defined in the introduction section, line 41: “ Latent Tuberculosis Infection (LTBI) is defined as a status of infection control due to a persistent immune response generated by the antigens of the M tuberculosis complex without evidence of clinical manifestations of active TB."
Q5. We thank to reviewer this suggestion. However, since we explain all the baseline characteristics in the results section, we considered that table 4 and 8 fits better in this section.
Q6. The conclusion has been modified as follows: “In our study, the indication of LTBI screening and treatment defined groups with different sociodemographic and clinical characteristics that defined different risks for AE and treatment outcome. We observed a higher risk of developing AEs in the group of persons receiving IS treatment and a lower adherence to treatment in recently arrived immigrants from countries with a high incidence of TB. Strategies should be strengthened to increase adherence to treatment in recently arrived immigrants and to ensure comprehensive follow-up of AE in those with baseline IS.”
